# ERRORAUG: MAKING ERRORS TO FIND ERRORS IN SEMANTIC SEGMENTATION

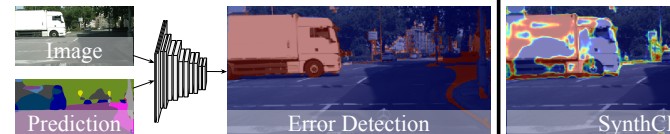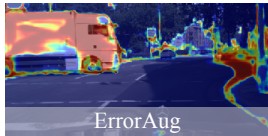

Figure 1: We propose ErrorAug as a simple and reliable approach for pixel-wise error detection. ErrorAug allows us to artificially generate more examples of errors which are also of a higher degree of difficulty. ErrorAug improves relative performance of key error detection metrics by over 7.8%/11.2% for in-domain/out-of-domain scenarios versus previous state-of-the-art approach SynthCP.

## ABSTRACT

In order to develop trustworthy downstream applications for semantic segmentation models, it is important to not only understand the performance of a model on datasets, but to localize areas where the model may produce errors. Pixel-wise error prediction of semantic segmentation maps is a challenging problem in which prior work relies on complicated image resynthesis pipelines. We introduce *error augmentation, a framework which enables us to learn robust error detectors by applying data transformations independently on the predicted segmentation maps. This approach enables direct prediction of pixel-wise error in semantic segmentation maps, an approach explored as a naive baseline in prior works, to achieve state of the art performance. As a proof-of-concept we propose a series of three simple transformations that generate challenging segmentation errors by swapping pixel predictions within a segmentation map. Our approach outperforms previous methods of error detection for semantic segmentation across all metrics and improves performance by over 7.8% on AUPR-Error. Additionally, we show that our approach not only generalizes to unseen test examples, but remains reliable despite significant shifts in the target domain.*

## 1 INTRODUCTION

Understanding when machine learning models are producing inaccurate predictions is essential for improving the reliability of systems that build upon these models. Recent works in performance prediction have made strides in predicting the performance of classification systems in novel environments Garg et al. (2022); Chen et al. (2021); Guillory et al. (2021). However for complex computer vision tasks like semantic segmentation, its important to not only identify when a model produces and inaccurate prediction but also where the models predictions have failed. For instance, in a robotics setting a misclassification far away from the action space of the robot may be less relevant than one in the immediate pathway. As such the task of pixel-wise error detection becomes increasingly important as we strive to produce AI systems that can safely interact with ever-changing real world environments.

We propose Error Augmentation (*ErrorAug*), a process for synthesizing challenging localization errors by applying data transformations on predicted class probabilities independent of any transformations on the input images, as a step for training high-quality pixel-wise error detectors. In order to demonstrate the effectiveness of this process, we propose three swapping operations that when applied to a segmentation map allow us to treat error detection as a supervised learning task and

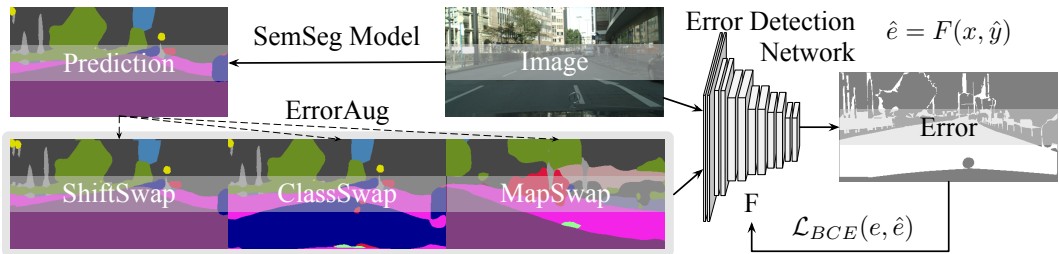

Figure 2: Error augmentation allows use to directly train an error detection network with supervised learning. Applying ErrorAug over the model's prediction allows us to create a challenging and diverse set of example errors. Training according to this pipeline leads to error detection model which performs reliably on novel examples.

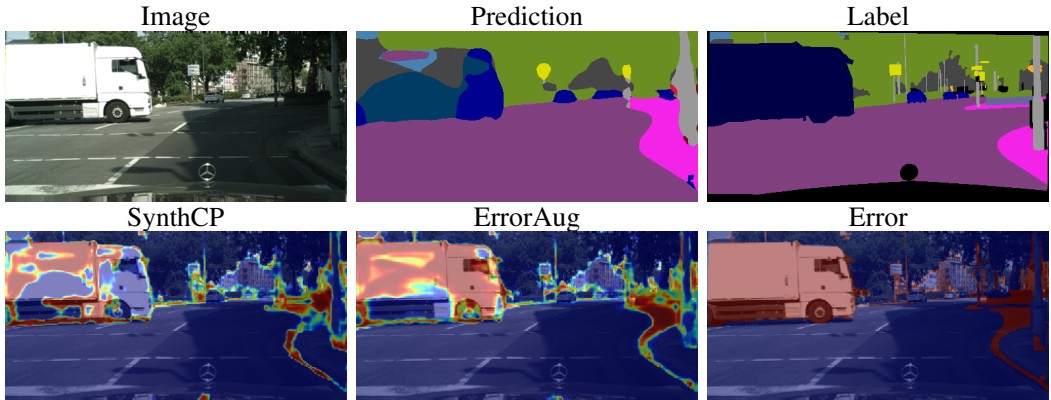

Figure 3: The top row illustrates a semantic segmentation task where a segmentation map is predicted from an image. The difference between the top row prediction and the top row label is the error map, depicted in the bottom right. The goal of our approach, ErrorAug, is to accurately predict the error map on the bottom right, based on the image and the prediction. We see that despite the model's simplicity, ErrorAug does a much better job at locating large misclassified regions, like the moving truck in this example, than the previous state-of-the-art approach SynthCP.

directly train a model to predict pixel-wise error maps for semantic segmentation systems. Most prior works in this space have attempted to directly train error detectors , but due to their tendency to overfit in this problem setting, they are most frequently presented as naive baselines to motivate more complicated approaches. In this work we show that by including our *ErrorAug* process into previous pipelines we bypass the complications of previous approaches and can produce state-of-the-art results using prior codebases direct prediction implementations. Our training procedure is illustrated in figure 2.

The most prominent prior work in this space built upon the success of conditional generative models by using the predicted segmentation map to condition a model whose goal it is the recreate the input image Xia et al. (2020)Di Biase et al. (2021). The discrepancies between this reconstruction and the original input image are then used to detect errors. These approaches are not only expensive, but are overly-sensitive to the training dataset as the behavior of these generative models on changing domains remains or poorly understood. Our approach does not require complicated auxiliary tasks or networks. We dramatically improve upon state-of-the-art approach, SynthCP, by simplifying the modeling pipeline into a single neural network, optimizing with a standard binary cross entropy loss, and extending the set of errors observed in training. Figure 1 and 3 illustrate the apple-to-apple comparison between SynthCP (SOTA) and ErrorAug (ours).

Error Augmentation, in the context of error detection, can be used to describe any data transformation applied independently to the label predictions in the training pipeline of an error detector that allows the derivation of novel target error maps. In this work, we explore three swapping data transformations (*ShiftSwap*, *ClassSwap*, and *MapSwap*) which generate different types of challenging

errors. *ShiftSwap* involves a spatial shifting of predictions in a segmentation map and thus induces errors over areas where class boundaries exist. *ClassSwap* involves a swapping of the prediction probabilities between classes in an image, inducing misclassification errors. The last swapping operation is meant to induce errors of all types, as it involves swapping the predicted segmentation map of one image with that of another image, *MapSwap*. Figure 4 illustrates all these operations over the segmentation maps of a single image. While some of the changes are drastic and immediately noticeable, others are more subtle but still contribute novel examples for training.

Our work shows that each of these shifts alone is powerful enough to learn a direct prediction model which outperforms prior art. When all shifts are introduced to the same training pipeline we show that we are able to train a model that improves on prior art by a margin of $> 7\%$ on error detection on the Cityscapes Cordts et al. (2016) benchmark. More importantly, we demonstrate that this approach is considerably more robust when dealing with instances of domain shift, as demonstrated by outperforming image re-synthesis approaches by $> 13\%$ when models trained over the European benchmark of Cityscapes are evaluated over the largely North American dataset, BDD100K Yu et al. (2020).

## 2 RELATED WORK

We focus on the task of pixel-wise error detection for semantic segmentation which can be described as identifying erroneously label pixels over objects with known classes. Anomaly detection for semantic segmentation is a closely related task that instead focuses on identifying pixel of objects with unknown classes Di Biase et al. (2021). Some approaches in prior literature attempt propose solutions to both of these tasks, we instead focus on the pixel-wise error detection tasks and also extend our analysis to error-detection performance over domain shifts. A popular baseline in this space is MSP-CRF Hendrycks et al. (2019a), which extends the maximum softmax probability method Hendrycks & Gimpel (2016) for outlier detection. In order to better account for the spatial element of pixel-wise error detection, they chose to include a conditional random field Lafferty et al. (2001) component which can better account for locality. Other predictive uncertainty-based approaches such as MC-Dropout Gal & Ghahramani (2016), deep ensembles Lakshminarayanan et al. (2017), and Bayesian SegNet Kendall et al. (2015) have been explored for pixel-wise error detection, but fail to consistently detect misclassification.

More recent works in this space have used conditional generative models to attempt to resynthesize the original image based on the predicted segmentation map Di Biase et al. (2021); Lis et al. (2019); Xia et al. (2020). A separate module is then developed to determine where the resynthesized image and the original image substantially differ and these regions are estimated as likely containing erroneous predictions. These approaches rely on complicated auxiliary tasks, such as image reconstruction, in order to prevent their error detection model from becoming overly reliant of the predicted probabilities generated by the segmentation model. The approach introduce a substantial amount of computational overhead, and produce methods which do not generalize well to novel domains.

By introducing error augmentation as a framework for learning to directly predict error maps for semantic segmentation, we forgo the issues of predictive uncertainty methods which are too reliant on model confidences and do not properly account of spatial structure. As our error augmentation approach is not dependent on a generative model, we are also able to avoid the computational complexity and sensitivity to domain shift observed in image resynthesis approaches.

Our additional focus on evaluating under domain shift stems from several lines of recent works showing that existing model architectures struggle to maintain consistent performance across data shifts where existing model architectures struggle to maintain consistent performance Recht et al. (2019); Koh et al. (2021); Madras & Zemel (2021); Peng et al. (2018); Bashkirova et al. (2021). Several works have demonstrated the ability to predict model performance in these new environments Deng & Zheng (2021); Guillory et al. (2021); Rabanser et al. (2019); Garg et al. (2022); Chen et al. (2021), however these approaches do not readily translate to the complex task of pixel-wise error detection.

Outside of error detection, the machine learning community has experienced an explosion in research concerned with various forms of data augmentation Shorten & Khoshgoftaar (2019) of which

error augmentation can be viewed as a special case. Data augmentations have been central to the development of state-of-the-art methods in self-supervised learning Doersch et al. (2015); Misra & Maaten (2020), semi-supervised learning Sohn et al. (2020), domain adaptation Sun et al. (2020), and uncertainty quantification Hendrycks et al. (2019b). Data augmentation strategies have primarily been deployed in one of two ways in prior work. First is as a method of insuring model consistency with respect to certain data transformations. Augmentation strategies which aim for model consistency must take care to ensure that the data transformations they expose the model to do not alter the underlying task. Other approaches use data augmentations to learn auxiliary tasks. Learning over well-motivated auxiliary tasks can lead to improved downstream model performance. The data transformations in error augmentation are only applied to one of the inputs and their application changes the target labels in a non-symmetric manner. This produces an approach in which model consistency is not the goal of the augmentations since they lead to a changing set of labels, and the *auxiliary* task produced by the labels happens to be the same task as the primary task.

## 3 ERROR AUGMENTATION

In this section we introduce Error Augmentation, ErrorAug, as a framework for learning more reliable error detectors for semantic segmentation. ErrorAug contributes two new components to the training pipeline for direct error prediction, synthetic segmentation maps and automatically derived error maps.

We first discuss the formulation and motivation of the proposed error detection framework, ErrorAug. Next, we describe three types of error augmentations (ShiftSwap, ClassSwap, MapSwap), all of which are quick and easy to implement. Lastly, wediscuss the neural network architectures used to represent the error detectors that we train and how they differ from prior approaches.

### 3.1 ERROR SYNTHESIS

Our semantic segmentation task consist of $L$ distinct semantic labels. An image $x$ of size $w \times h$ could be labeled with a semantic segmentation map $y \in \{0,1\}^{L \times w \times h}$ A semantic segmentation model $P$, works by operating over image $x$ to produce a estimate of the semantic segmentation map $P(x) = \hat{y} \in \{0,1\}^{L \times w \times h}$. We define an error map $e \in \{0,1\}^{w \times h}$ such that :

$$e_{i,j} = \max_{0 \leq l \leq L} y_{l,i,j} \neq \hat{y}_{l,i,j} \tag{1}$$

The goal of error detection algorithms is to learn a model $F$ such that the error map $e$ can be estimated with $F$ from the image $x$ and predicted segmentation map $\hat{y}$, thus $\hat{e} = F(x, \hat{y})$, as described in algorithm 1. Error detection can be formulated as a supervised learning problem where the target labels are error maps $e$ and the input instances are images $x$ and predicted segmentation maps $\hat{y}$. Letting $\mathcal{L}$ represent a loss function, such as binary cross entropy, our error detector $F$ can be learned as a minimization of the following equation:

$$\min_{F} \mathcal{L}_{\text{BCE}}(F(x, \hat{y}), e) \tag{2}$$

Following standard procedure in supervised learning, datasets are split into a training subset and a test subset with the test subset being used to assess the models performance on previously unseen examples.

Due to high levels of correlation between the segmentation map $\hat{y}$ and the error map $e$, it is possible to learn direct error detectors $F$ which perform well on the training set, with little comprehension about the image $x$ or the label space $\mathbb{L}$. Prior work uses the auxiliary task of image synthesis to encourage the error detection model to learn non-trival information about the images and/or the segmentation task. Instead of synthesizing images, our framework aims to ensure the error detection model is based on a sufficient understanding of the segmentation task by synthesizing challenging examples with less correlation between the segmentation map $\hat{y}$ and the error map $e$.

We define Error Augmentation as the process of applying a data transformation $T$ to the predicted segmentation map $\hat{y}$ such that it generates a new predicted segmentation map $\hat{y}' = T(\hat{y})$. Unlike traditional data augmentation strategies, the transformation $T$ is intentionally not applied to the input

| Prediction | ShiftSwap | ClassSwap | MapSwap | Label |

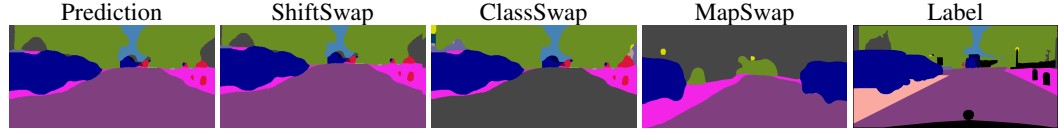

Figure 4: Error augmentation works by applying data transforms directly on the prediction of a segmentation model. Illustrated above, ShiftSwap, ClassSwap, and MapSwap are the data transforms we propose for error augmentation. By comparing the prediction and transformed predictions with the target label map, on the far right, we can see how each transform contribute unique errors to the training set of our error detector.

image $x$. As such, it is not enough for model $P$ to be equivariant to transformation $T$, and the error detection model $F$ must attempt recover the relationship between $\hat{y}'$ and the input image $x$. From the transformed segmentation map $\hat{y}'$ and the known segmentation map $y$, a new error map $e'$ can be derived:

$$e'_{i,j} = \max_{0 \leq l \leq L} y_{l,i,j} \neq \hat{y}'_{l,i,j} \tag{3}$$

The relationship between $T$ and the new error map $e'$ is dependent upon the ground truth label map $y$, and can be complex even for simple data transforms. In traditional data augmentation settings, care must be taken when choosing the set of data transforms as certain transformations may alter the relationship between the transformed image $x' \leftarrow T(x)$ and observed label $y$. For instance, synthetically changing the colors of an observed snake, may lead to a meaningful change in classification of the observed animal. However, since our input image remains unaltered, as long as the transformed map $\hat{y}'$ is a valid segmentation map, the derived error map $e'$ is well-defined. We will take advantage of this in later sections by proposing transforms that would be nonsensical if applied to the input image. This formulation allows us to rewrite the minimization objective as operating over $\hat{y}'$ and $e'$ and allows us to arbitrarily scale the number and complexity of training examples by adjust our transformation function $T$.

$$\min_F \mathcal{L}_{BCE}(F(x, \hat{y}'), e') \tag{4}$$

The quality of the learned error detector, $F$, will be largely dependent upon the nature of the transformation function $T$ and the architecture of the model. In this work we explored three relatively straightforward data transformations and observe they all have a strong positive impact, motivating future work exploration of error augmentation strategies and their relationship with robust machine learning.

## 3.2 Swapping Transformations for Error Detection

In order to better understand the utility of our proposed framework for Error Augmentation, we implemented three swapping data transformations (ShiftSwap, ClassSwap, MapSwap) which increase the expected delta $\delta$ between the predicted segmentation map $\hat{y}$ and the transformed map $\hat{y}'$

$$\delta = \|\hat{y} - \hat{y}'\| \tag{5}$$

Our proposed transformations where to designed to generate specific forms of errors, such as misclassifications and errors at object boundaries. In figure **??**, we visualize these these three transformations over the same image, illustrating how the prediction maps change and the subsequent new error maps that result from the transformations. For our learned model, $ErrorAug$, we stochastically compose all three transformations. In ablation studies, we investigate the contribution of each one independently.

**ShiftSwap** is a positional swapping operation which selects a spatial window of the predicted label map $\hat{y}$ and shifts this window by certain horizontal $\delta_x$ and vertical $\delta_y$ directions, overwriting all previous predictions in this new location. For all following swapping operations, we assume that $\hat{y}'$ is initialized with the values from $\hat{y}$. Given $a, b$ and $c, d$ as the horizontal and vertical boundaries for the spatial window and assuming that all indices are valid, this operation could be represented as follows:

$$\hat{y}'_{:,a+\delta_x:b+\delta_x,c+\delta_y:d+\delta_y} \leftarrow \hat{y}_{:,a:b,c:d} \tag{6}$$

---

**Algorithm 1** ErrorAug: Making Errors to Find Errors

---

Given dataset $D$, semantic segmentation model $P$, error detection network $F$
**for** input batch $(x^B, y^B) \in D$ **do**         ▷ Operate over images $x^B$ and labels $y^B$
    $\hat{y}^B \leftarrow \text{ErrorAug}(P(x^B))$            ▷ ShiftSwap, ClassSwap, MapSwap
    $e^B \leftarrow \mathbb{1}(y^B \neq \hat{y}^B)$                 ▷ Derive error map
    $\hat{e}^B \leftarrow F(x^B, \hat{y}^B)$                  ▷ Predict error map
    Optimize network $F$ by back-propagating $\mathcal{L}_{BCE}(e^B, \hat{e}^B)$
**end for**

---

This approach results in shifted prediction boundaries and encourages the error detection module to learn the correct boundary of positions.

**ClassSwap** is a semantic swapping operation which arbitrarily chooses two semantic classes in a label map and swaps their predictions throughout at each pixel location in the label map. Given classes $p$ and $q$ as the intended swapping pair, the ClassSwap operation could be represented as follows:

$$\hat{y}'_{p,:,:} \leftarrow \hat{y}_{q,:,:}; \tag{7}$$

$$\hat{y}'_{q,:,:} \leftarrow \hat{y}_{p,:,:} \tag{8}$$

Since this approach respects the spatial predictions of the model $P$ it may result in regions with correct segmentation boundaries but incorrect semantic classes.

**MapSwap** is the last and most drastic transformation we propose. It is designed to swap the predicted pixel-level label maps between images. Letting $x^{(i)}$ and $x^{(j)}$ represent two distinct images and $\hat{y}^{(i)}$ and $\hat{y}^{(j)}$ represent the predicted segmentation maps of these images, respectively. Then MapSwap operation could be represented as follows:

$$\hat{y}'^{(i)} \leftarrow \hat{y}^{(j)}; \tag{9}$$

$$\hat{y}'^{(j)} \leftarrow \hat{y}^{(i)} \tag{10}$$

This results in segmentation maps which are not derived from the input image and thus encode no information about the segmentation models properties over this particular input image $P(x)$. While this approach has severed the causal relationship between the input image $x$ and the synthetic segmentation map $\hat{y}'$, it is still likely to produce an informative error map $e'$, which is neither all ones or all zeros due to structural similarities between scenes. These structural similarities such as roads being likely to occur near the bottom an image and the sky being likely to be near the top of an image are illustrated in Figure 4 or through more examples in the supplementary materials. This is an example of a data transformation which would be degenerative if applied over the input image $x$ but yields challenge and useful information when employed for error detection.

## 4 EXPERIMENTS

### 4.1 DATASET

We intensively evaluate our method on two semantic segmentation benchmarks for autonomous driving, Cityscapes Cordts et al. (2016) and BDD100K Yu et al. (2020).

**Cityscapes** With 2975 high-resolution training images and 500 validation, the Cityscapes dataset, is a large-scale semantic segmentation datasets which prior works have used to validate the quality of error detectors. The dataset consist of images captured from vehicles in various European cities for the task of improving visual systems for self-driving. The pre-defined train IDs contains 19 semantic classes, and also contains out-of-context labeling upon which errors and accuracy are not counted.

**BDD100K** We look at the semantic segmentation setting for the BDD100K dataset, which consist of 7000 training images, and 1000 validation images for semantic segmentation. The dataset consist

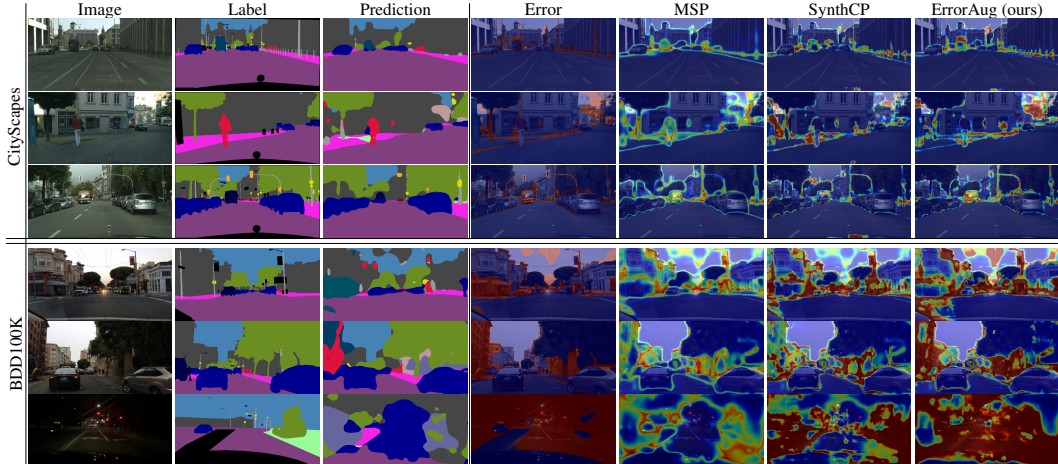

Figure 5: MSP, SynthCP, and our ErrorAug are representative algorithms for approaches based on predictive uncertainty, image resynthesis, and direct prediction. We visualize the predicted error maps and the actual error maps for each of these algorithms across multiple images from both the in-distribution Cityscapes dataset and BDD100k which represented a shifted target domain. ErrorAug is able to outperform SynthCP and MSP across both domains and the results above illustrate how this improvement is noticeable on the more challenging setting of error detection over domain shift.

of images captured from vehicles throughout cities, highways, and rural landscape in North America. In addition to supporting the same 19 semantic classes as Cityscapes, it also contains various attribute annotations for the images, such as weather conditions and environmental settings. As we are interested in how our models perform when exposed to novel environmental settings we evaluate the quality of our error detectors trained on Cityscapes in the BDD100K dataset.

## 4.2 SETUP

**Implementation**  We title our error detection model as *ErrorAug* because it is trained over the proposed set of swapping error augmentations. To build our training dataset for ErrorAug follow the formulation used in Xia et al. (2020), where they use four-fold cross validation in order to address the smaller size of the validation dataset and ensure there exists an held-out set not seen by either the segmentation model or the error detection model. The held-out set for each fourfold cross validation step is annotated with the prediction probabilities for the segmentation model trained at this step. The final error detector can then be validated over the held-out test set and an error detector and segmentation model both trained over the entirety of the training set. The ErrorAug transformations used in our model are the composed set our three swapping operators (ShiftSwap, ClassSwap, MapSwap), each implement with a $p = \frac{1}{3}$ chance of being activated.

**Architecture**  Since our approach presents a more challenging error detection problem than those explored in previous works, we require an error detection architecture with sufficient capacity. Our architecture consist of two ResNet-18 He et al. (2016) branches, one for the image input $x$ and one for the predicted label map $\hat{y}$. The image ResNet is initialized by ImageNet-pretrained weights, while 19-channel segmentation map ResNet is trained from scratch with Kaiming Initialization He et al. (2015). Features are extracted from the siamese neural networks, concatenated with each other and then upsampled to allow for direct pixel-wise prediction of errors. Prior direct prediction approaches concatenated the segmentation map and image prior to passing it through a neural network, we observe that this approach is unable to adequate capture the complex relation between the segmentation map and input image required to learn an adequate detector under the ErrorAug framework, as shown in the ablation experiments in table 3.

Table 1: Experiments on Cityscape Dataset. We detect pixel-wise failures on the results of FCN-8 and Deeplab-v2. We report our results of ErrorAug and several baselines error detectors. ErrorAug outperforms all prior approaches across all observed metrics, indicating that when properly trained direct prediction strategies are a promising underexplored approach for error detection. ErrorAug offers $> 7.5\%$ relative improvement on AUPR-Error over the more computationally intensive prior state-of-the-art SynthCP for both models explored.

| Method | FCN-8 | | | | | Deeplab-v2 | | | | |
| | Ap-Err ↑ | Ap-Suc ↑ | AUC ↑ | FPR95 ↓ | mIoU ↑ | Ap-Err ↑ | Ap-Suc ↑ | AUC ↑ | FPR95 ↓ | mIoU ↑ |
|---|---|---|---|---|---|---|---|---|---|---|
| MSP | 50.31 | 99.02 | 91.54 | 25.34 | | 48.36 | 99.12 | 91.68 | 27.27 | |
| MCDropout | 49.23 | 99.02 | 91.47 | 25.16 | | 47.85 | 99.23 | 92.19 | 24.68 | |
| TCP | 48.54 | 98.82 | 90.29 | 32.21 | 51.15 | 45.57 | 98.84 | 89.14 | 36.98 | 58.27 |
| SynthCP | 55.53 | 99.18 | 92.92 | 22.47 | | 49.99 | 99.34 | 92.98 | 21.69 | |
| ErrorAug (ours) | **59.84** | **99.31** | **93.94** | **20.23** | | **53.84** | **99.40** | **93.77** | **20.38** | |

Table 2: Experiments on BDD Dataset which models and Error Detectors trained on Cityscapes. We detect pixel-wise failures on the results of FCN-8 . ErrorAug outperforms all prior approaches across all observed metrics and offers a $13.6\%$ relative improvement on AUPR-Error over the more computationally intensive prior state-of-the-art SynthCP. An interesting observation is that the much less sophisticated MSP approach outperforms SynthCP on our primary metric AUPR-Error over BDD Dataset.

| Method | Ap-Err↑ | Ap-Suc ↑ | AUC ↑ | FPR95 ↓ | mIoU |
|---|---|---|---|---|---|
| MSP | 59.05 | 94.79 | 84.52 | 42.07 | |
| SynthCP | 58.26 | 95.34 | 85.34 | 39.69 | 25.82 |
| ErrorAug (ours) | **66.21** | **96.49** | **88.98** | **34.58** | |

**Baselines** To replicate prior work, we choose FCN-8s Shelhamer et al. (2016) and Deeplab-V2 Chen et al. (2017) as our default semantic segmentation models to predict the error map. The primarily goal of our work is to improve the capability of performing pixel-level error map prediction for the given segmentation model. We compare the proposed ErrorAug with MSP Hendrycks & Gimpel (2016), MCDropout Gal & Ghahramani (2016), TCP Corbière et al. (2019), and SynthCP Xia et al. (2020). Following the experimental setup of SynthCP, we also implement a fairly comparable baseline, leveraging regression head to directly predict the pixel-level failures given both input image and the corresponding predicted label map. All these baselines share the same backbone and training strategies. We further ablate the sensitivity of some hyper-parameters in the proposed ErrorAug system.

**Metrics** In order to evaluate the effectiveness of our approaches, we use the following four metrics: AUPR-Error (denoted as Ap-Err, higher is better), AUPR-Success (denoted as Ap-Suc, higher is better), AUROC (denoted as AUC, higher is better), and FPR at 95% TPR (denoted as FPR95, lower is better) as in Liu et al. (2019) and Xia et al. (2020). The main metric we concern ourselves with in this work is AUPR-Error, which computes the area under the precision-recall curve by treating errors as the positive class.

## 4.3 COMPARISON WITH STATE-OF-THE-ART METHODS

**Cityscapes** In table 1, we observe that $ErrorAug$ substantially outperforms prior methods over all observed metrics. Improving relative performance on the main metric of AUPR-Error by $> 7.5\%$, we show that this model can outperform models with considerably more parameters and a more complex evaluation cycle. In figure 5, we visualize some of the results and observe that our method does

Table 3: Ablation Experiments on Cityscape Dataset. We evaluate each proposed swapping operation in our approach as well as a variant which uses the proposed architecture to directly predict errors with no swapping. We highlight results which outperform previous state-of-the-art methods and show that each variant of our approach independently outperforms prior methods, with *Map-Swap* providing the largest improvement. Without error augmentation our proposed architecture still out performs other direct prediction methods such as that found the Old Arch from Xia et al. (2020). When exposed to the full set of error augmentations, the performance of the prior architecture suffers due to its limited capacity.

| Method | Ap-Err ↑ | Ap-Suc ↑ | AUC ↑ | FPR95 ↓ |
|---|---|---|---|---|
| Old Arch - NoAug | 52.17 | 99.15 | 92.55 | 22.34 |
| ErrorAug-Old Arch | 54.31 | 99.18 | 92.85 | 21.91 |
| ErrorAug-NoAug | 57.63 | 99.14 | 92.75 | 23.72 |
| ErrorAug-MisClass | 58.50 | 99.21 | 93.30 | 21.69 |
| ErrorAug-Shift | 58.83 | 99.21 | 93.25 | 21.78 |
| ErrorAug-Image | 59.39 | 99.30 | 93.80 | 20.86 |
| ErrorAug | **59.84** | **99.31** | **93.94** | **20.23** |

a better job at identifying region-level misclassifications that other approaches. Furthermore, we note that this ability becomes more important as contexts shift such as in the *BDD100K* examples.

**BDD100K** In table 2, we compare the performance of error detectors on data obtained from a new domain (train on Cityscapes, test on BDD100K). We observe an even larger relative improvement of 13.6% in our primary metric AUPR-Error. Furthermore, we observed that in continues to outperform all other approaches in all four of the metrics. An interesting observation, is that parameter-free MSP approach outperforms the expensive SynthCP approach in this new domain over our primary metric of AUPR-Error. In the supplemental material we visualize the results of these approaches.

### 4.4 ABLATION STUDY OF KEY COMPONENTS

In order to understand the impact of different elements of our approach we run ablation experiments as reported in table 3. We note that an error detector trained on any of the 3 swapping operations proposed in this paper would outperform all prior work across all reported metrics. Additional we observe, that while our new error detection architecture would outperform prior approaches over AUPR-Error without using error augmentation. However, it would not improve across all metrics and it's performance is significantly increased when paired with our challenging error augmentation settings. Lastly, we see that when attempting to train a direct prediction model which concatenates the segmentation map with the image before input, the model performs worse that if trained without error augmentation. This result indicates the importance of high-capacity models to solve this more challenging direct prediction task.

## 5 CONCLUSIONS

We develop a framework for learning high-quality and robust error detectors for semantic segmentation. Our ErrorAug method performs independent data transformations over predicted semantic segmentation maps and is able to generate arbitrarily many novel and challenging error detection tasks. Using this to directly learn an error detection model leads to state-of-the-art error detection results for each swapping transformation we explored. ErrorAug produces substantially better error detection models with a significantly simpler approach than prior methods. Each augmentation operation we explored has consistently improved results, yet we have only investigated the three swap operations reported above. We believe there is ample opportunity for future exploration of further improved data transformations and error detection architectures.

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
