# OpenReview forum: "ErrorAug: Making Errors to Find Errors in Semantic Segmentation"
_ICLR.cc/2023/Conference — Submitted to ICLR 2023_

### Official Review · Reviewer_iFWU · 2022-10-24

**Confidence:** 4
**Correctness:** 3
**Technical Novelty And Significance:** 2
**Empirical Novelty And Significance:** 3
**Recommendation:** 5

**Clarity, Quality, Novelty And Reproducibility:**

Both the clarity and quality of this work are good, but could be improved further (see a few points on this above).
The method should be straighforward to reproduce given its simplicity.
The technical novelty is rather modest, since it's an application of the classical data augmentation paradigm to a new domain. The empirical results showing that it's quite effective can be interesting to some audience interested in uncertainty estimation and out-of-domain generalisation.

**Strength And Weaknesses:**

Strenghts:

- The idea to use data augmentation is straightforward, easily implementable and seems to be more effective than previous works.
- I like that there are cross-domain experiments (training on Cityscapes, testing on BDD).
- There is a fair amount of qualitative results and comparisons which improve the presentation quality.

Weaknesses:

I’m not sure where to position this work.

On one hand, there are uncertainty estimation methods, such as MC-Dropout, that strive to provide calibrated confidence. One could binarise the confidence to obtain an “error detection” method. Although there is one comparison to MC-Dropout, the methodology of training this baseline and its architecture specifics (e.g. one needs to place Dropout layers carefully) are unclear. I am curious if the predictions of the error detection network can be used for uncertainty estimation. For example, how would expected calibration error compare to that of MC-Dropout? I also wonder how this approach compares to test-time augmentation, i.e. augmenting the input image and averaging the semantic maps to obtain calibrated confidence. These would be two off-the-shelf baselines.

On the other hand, there are anomaly detection methods that focus on identifying out-of-distribution phenomena in the test data. Here, the paper lacks appropriate comparisons as well, since these methods are typically evaluated on anomaly detection benchmarks, e.g. StreetHazards, which is missing in the experiments. Running the method on BDD after training on Cityscapes goes in this direction, but both are the datasets of traffic scenes, after all.

The notation could be improved (e.g. missing Iverson bracket in Eq. 1, 3)
The quality of presentation can be improved significantly. There are numerous typos (e.g. “examples by adjust our […]”, “widiscuss”, etc.), missing cross-references (e.g. Sec. 3.2).




**Summary Of The Paper:**

The work tackles the problem of error detection in semantic segmentation. The naive approach of training a binary pixel-wise classifier on the image and the corresponding semantic map (produced by a segmentation network) is extended by perturbing the semantic map with a three types of straightforward transformations. The experiments indicate that such a binary classifier would surpass previous best methods on standard benchmarks of error detection.

**Summary Of The Review:**

Overall, I like this work. However, there is lack of experimentation both in terms of uncertainty estimation and anomaly detection, which needs to be addressed.

**Post-rebuttal comment**

I thank the authors for their response. As before, I like the simplicity of this approach and the consistency improvement over the baseline.
However, I downgrade my rating as I do agree with the other reviewers that the experimentation is a bit too conservative:

* Missing MC-Dropout and TCP in the out-of-distribution setting [R-G1qP]
* The increase in model capacity is not discussed [R-G1qP]. In fact, as R-2JRb points out, Table 3 suggests that around half of the improvement comes from architectural changes (increasing the number of parameters) rather than from erro augmentation.
* Missing comparisong to FSNet. I am not sure this would be an apple-to-apple comparsion by just copying the values due to the differences in the segmentation accuracy and the validation set. These points should be discussed, nevertheless.
* There is not sufficient analysis of this work in the context of anomaly detection and/or uncertainty estimation, as typical in comparable works (e.g. in SynthCP).

---

> ### Author Response · Authors · 2022-11-19
> **Calibration and Anomaly Detection**
>
> The reviewer is right in identifying that this work in pixel-wise error detection is closely related to the two more frequently explored research questions of Anomaly Detection and Uncertainty Quantification and there is potential to expand Error Augmentation to include both of these areas. Expanding this Error Detection framework into anomaly detection would require a significant shift in our approach as it would require us to be able to synthetically generate anomalies instead of errors as the current method does. However expanding this work to produce uncertainty estimates is relatively straightforward and would likely require a calibration of our binary error detector.

---

### Official Review · Reviewer_2Ds6 · 2022-10-24

**Confidence:** 5
**Correctness:** 2
**Technical Novelty And Significance:** 2
**Empirical Novelty And Significance:** 2
**Recommendation:** 5

**Clarity, Quality, Novelty And Reproducibility:**

Clarity: Simple to understand despite somewhat sloppy presentation.

Quality and Novelty: Somewhat limited due to Point 4 in the above section.

Reproducibility: Most parts of the paper are readily reproducible.

**Strength And Weaknesses:**

Strengths:

1. The empirical observation that two separate encoding networks (one for the input image and the other for the prediction map) can boost the prediction performance is a plus.

Weaknesses:

1. The paper looks like a rush one with not-hard-to-spot typos on each page.

2. The reviewer is not a fan of augmentation, as it is generally not transferrable to unseen augmentations. To correct the reviewer's biased view, the authors are encouraged to conduct cross-model experiments (e.g., training on FCN-8s / DeepLab v2 and testing on other segmentation models).

3.  How to do boundary handling in ShiftSwap and how to control the similarity level between swapped map and the original one in MapSwap?

4. The primary motivation for developing a failure predictor, either global or local, is to spot hard examples (in an online fashion) to prioritize the training process or to identify catastrophic failures (in an offline fashion) to actively fine-tune the model for improved performance. These results however are not obtained in this paper.

**Summary Of The Paper:**

This paper introduces a set of data augmentation techniques for predicting dense error maps in semantic segmentation.

**Summary Of The Review:**

With better experimental designs (including prioritizing the training process and active fine-tuning) and more comprehensive evaluations (in cross-model and cross-dataset settings), this paper may have a chance to get in.

---

> ### Author Response · Authors · 2022-11-19
> **Cross model evaluation**
>
> (Unseen Augmentations)
> None of the models are evaluated over any augmentations, so the need to evaluate over unseen augmentations at test-time is not within the scope of the proposed approach or intended use cases.
> While the idea of evaluating across architectures is interesting, it amounts to redefining of the problem statement into "universal error detectors" and was not explored in this work.
>
> (Boundaries in ShiftSwap, Similarity in MapSwap)
> We do not control for similarity in MapSwap though reporting the average prediction overlap may be useful in further understanding these particular augmentation strategies. The boundary areas in ShiftSwap remain constant, thus some predictions are repeated.
>
> (Primary Motivation of Error Detection)
> We would argue that the primary purpose of error detection models is for decision making under uncertainty. Whether to abstain from action, retrain or actively fine-tune are all distinct possibilities but are tangential to the core task of detecting errors.

---

### Official Review · Reviewer_2JRb · 2022-10-27

**Confidence:** 3
**Correctness:** 3
**Technical Novelty And Significance:** 2
**Empirical Novelty And Significance:** 3
**Recommendation:** 5

**Clarity, Quality, Novelty And Reproducibility:**

Novelty:

The paper uses augmentation concepts that have proven valuable in semi-supervised learning for error detection. The contributions consist of mixing multiple augmentation strategies together and a different use of the architecture for error detection. While the latter looks very similar to SynthCP (ResNet-18), only the way the data is passed differs (concatenate RGB + prediction vs separate branch foreach - probably the biggest single insight of this paper).

Clarity:

The paper has only minor issues, but could use some help. E.g., Figure 2, please show somehow it's a dual branch network. I would shrink related work and add 3 rows from Fig 6 supp, it make the transforms clearer. Or make a mix of Fig 2+4+6, it would help understanding the common artifacts.

Not sure on how the testing on BDD100K was carried out, the whole training set was evaluated? FSNet claimed they picked a random 1k sample set.

Quality:

Although the problem and results are well described and mostly well reported, I believe the current pipeline is under-investigated. If a 6-year-old dual-branch NN with separate RGB / prediction yields half of the improvements, why not try a newer one? Why not concatenate the prediction augmentations? why not also cut the RGB image in random / semantic segmentation pieces? I would be very nice to see an edge label contamination discussion, similar to the ClassMix paper).

Reproducibility:

Most information is available in the paper for reproducing the results.

Misc

Minor paper issues, such as wediscuss (p4), Figure reference issue (p5),  uncertainity (p7)

**Strength And Weaknesses:**

### Strengths

- solid performance improvements over predecessor (SynthCP)
- straightforward error augmentation strategy - ShiftSwap, ClassSwap, and MapSwap involves shifting the image, swapping a class or the whole prediction
- fairly intuitive insight - if mixing unlabelled samples works for semi-supervised learning [1*] and domain adaptation [2*], variations of the method for error detection should also improve error detection performance
- can be added on top of any segmentation network

### Weaknesses

- generally poorer results compared to FSNet[3*] -- please cite and compare with
   - considering the (arguably most relevant metric) AUPR, the method yields 59.84 vs 67.83 on FCN8 ( _+22.15%_ for FSNet compared to +7.8% ErrorAug vs SynthCP) and 53.84 vs 57.84 on DeepLabv2 ( _+15.70%_ for FSNet compared to +7.7% ErrorAug vs SynthCP)
- ~50% of improvements are due to the architecture input change
  - even _without any augmentation_ , we are at ~+3.7 over SynthCP -- this means ~half of the error reduction is due to the different way of training the architecture, and only ~4% are due to the augmentation itself
  - I would speculate that a more powerful architecture and more augmentations would yield better results; nevertheless, the ablation study is frugal (I would add tick marks for each augmentation, I believe they are incremental?) and there are not many options explored (e.g., what happens if all three labels are concatenated, instead of the 1/3 chance of picking one)



___
[1*]Tranheden, W., Olsson, V., Pinto, J., & Svensson, L. (2021). Dacs: Domain adaptation via cross-domain mixed sampling. In Proceedings of the IEEE/CVF Winter Conference on Applications of Computer Vision (pp. 1379-1389).

[2*]Olsson, V., Tranheden, W., Pinto, J., & Svensson, L. (2021). Classmix: Segmentation-based data augmentation for semi-supervised learning. In Proceedings of the IEEE/CVF Winter Conference on Applications of Computer Vision (pp. 1369-1378).

[3*] Rahman, Q. M., Sünderhauf, N., Corke, P., & Dayoub, F. (2022). Fsnet: A failure detection framework for semantic segmentation. IEEE Robotics and Automation Letters, 7(2), 3030-3037.

**Summary Of The Paper:**

The paper proposes an error augmentation neural network to identify the errors made by semantic segmentation models by manipulating the predicted labels fed to the error detection network.

Three transforms are tested - shifting the prediction, swapping a class or completely changing the prediction labels. Along with the RGB image, one of these augmented prediction labels is selected at random and fed to dual-branch error detection NN that yields the final error map.

The authors claim state-of-the-art performance, with +~7.7 % AUPR error (49.99 vs 53.84) on Cityscapes in-distribution results. Furthermore, even out-of-distribution tests (BDD100K) yield +13.6 % AUPR improvement.

**Summary Of The Review:**

Although the paper yields good results on error detection from semantic segmentation, a (mostly) better performing / arguably SOTA method is left out. Disregarding the omission, I believe the augmentation techniques are under-exploited and could benefit from a better investigation. Otherwise, I believe the novelty is fairly limited, since shifting/pasting other labels have been tried and tested for better/domain adapted semantic segmentation.

---

> ### Author Response · Authors · 2022-11-19
> **Architectural improvements and Augmentation ablation.**
>
> (Architectural Improvements)
> The reviewers are correct in identifying a significant portion of the model improvement comes from the architectural changes and this could be better highlighted in the work, it is worth noting that simply using the direct prediction architecture which is not well suited to capturing changes in the label space, our ErrorAug approach still achieves results comparable to the SynthCP, as shown in table 3.
>
> (Ablation study + Concatenation)
> For clarification, each augmentation as 1/3 chance of being selected regardless of which other augmentations are selected. The process is compositional meaning multiple may be applied to any one image.
>
> (FSNet)
> We'd like to thank the reviewer for identifying this more recent related work. While we were unable to find code to reproduce this this work and it involves a change in the underlying learned models we're unable to directly apply ErrorAug to the models learned from this approach. However more in-depth discussion of this work in related works as well as inclusion of its now State-of-art results in our tables are merited.
>
> (BDD eval)
> We evaluate over the entirety of the bdd100k segmenation validation set.
>
> (Improved Augmentation approaches)
> We believe there are multiple better and stronger augmentation schemes than can further improve the results of and Error Detection network trained in an ErrorAugmentation fashion. This work represents a first and incredibly simple pass which shows that bringing data augmentation into this regime can enable direct prediction networks to work in a fashion not otherwise possible.

---

> > ### Comment · Reviewer_2JRb · 2022-12-02
> > **Better numbers within reach**
> >
> > Dear authors,
> >
> >
> > Thank you for your response!
> >
> > (Architectural~) I would still argue that this submission could have been significantly better with little effort. Why not start with pretrained, as reviewer G1qP suggested? This would have improved my rating if the numbers would have been provided.
> >
> > (Alblation ~) I see. Again, I believe a better separation would have helped to check which helps the most.
> >
> > (FSNet) I was thinking only of adding the numbers in the table, I don't think they will release the code.
> >
> > (Improved ~) I'm not saying this is a bad paper, only that a more significant jump in performance could have been achieved.
> >
> > All in all, I would like to keep my rating. Small changes could have made a significant difference, but for an unknown reason, the authors avoided them. Nevertheless, this is a promising research direction and I'm positive that it will get published in the future with the right experiment mix.
> >
> >
> >
> > All the best,
> > 2JRb

---

### Official Review · Reviewer_G1qP · 2022-11-01

**Confidence:** 4
**Correctness:** 3
**Technical Novelty And Significance:** 3
**Empirical Novelty And Significance:** 3
**Recommendation:** 5

**Clarity, Quality, Novelty And Reproducibility:**

- I perceive the proposed method as an improved baseline method for future work on error prediction of semantic segmentation methods, which is an important contribution
- To the best of my knowledge, I believe the idea is novel

There are a few writing and clarity issues that a revision would easily solve
- At the end of page 3, the "existing model architectures struggle to maintain consistent performance" is repeated twice
- Eq. 1,2 and Eq. 3,4 are pretty much the same and I believe the data augmentation could be folded into the equation to make it clearer, rather than leaving the definition of $e'$ and $\hat{y}'$ in text
- End of page 5 has a missing figure reference
- It is unclear which model was used for the BDD transfer experiment
- Training details are completely missing in the paper, which is a big detriment to reproducibility

**Strength And Weaknesses:**

# Strengths

- The simplicity of the proposed method over SynthCP that requires image synthesis in the loop is a big strength
- The proposed error augmentations are sound and well motivated. They cover class prediction errors and segmentation errors. The proposed map-swap augmentation smartly takes advantage of the fact that autonomous driving scenes often have correlated scene structure
- While the paper does not suggest this, error augmentation cou

# Weaknesses

- Evaluation follows SynthCP (from 2020) and only uses FCN-8s and Deeplab-v2 as the base segmentation models. At this point in time, these are far from state of the art models, which make much more fine-grained errors in Cityscapes semantic segmentation. It'd be important for the community to understand whether ErrorAug can be used to recover these finer errors in state of the art models such as ViT-Adapter

- The proposed siamese architecture for direct prediction uses double the parameters from the "OldArch" baseline (the naming convention could be much better here). It is therefore unclear if it's the siamese architecture or the increased parameter count that improved error prediction performance. A baseline for OldArch with comparable parameters would be ideal

- How does MCDropout and TCP do on BDD transfer? The transfer results do not show these

- The paper compares with SynthCP as is, which uses SPADE (CVPR 2019) as its conditional image synthesis backbone. In the 3 years since, there have been multiple improvements in image synthesis and it is entirely possible that a better conditional image synthesis backbone improves SynthCP's performance considerably

**Summary Of The Paper:**

ErrorAug proposes to use data augmentation for direct prediction of errors in semantic segmentation models. By augmenting errors in a model's predictions by swapping classes, shifting predicted mask or swapping segmentation from a different image, ErrorAug is able to train a more robust direct error prediction model. This is shown by showing better transfer of error prediction from Cityscapes to BDD100K, which has much more diverse imagery along with a domain shift from Europe to North America. The paper proposes to use a siamese architecture for direct error prediction instead of concatenating image and label in the channel dimension, which already beats synthesize and compare (SynthCP, ECCV 2020) on area under the precision-recall curve (Ap-Err, error prediction is treated as binary classification) by ~2 points for an FCN-8s model. The proposed data augmentation strategies additionally improve Ap-Err by ~2 points on Cityscapes semantic segmentation using FCN-8s. The paper also shows results using Deeplab-v2 as the base segmentation model.

**Summary Of The Review:**

Overall, ErrorAug shows that data augmentation can make direct error prediction from a model achieve better results than the state of the art method SynthCP, which synthesizes images conditioned on a predicted segmentation map and uses a learned comparison module to predict pixel-wise errors (and detect anomalies / predict IoU, which ErrorAug does not focus on) in a segmentation prediction. This should serve as an improved baseline and help the sub-community move forward.

However, ErrorAug is evaluated with at this point quite outdated semantic segmentation models and it is unclear whether state of the art segmentation model errors could be predicted well with these models. Moreover, due to the progress in conditional image synthesis, it is entirely possible that SynthCP, when implemented with newer image synthesis techniques would improve its performance.

Due to this, even though I believe the contribution and idea are valuable, I cannot recommend acceptance of the paper in its current form since the findings from the paper might not hold up when using state of the art methods from 2022.

---

> ### Author Response · Authors · 2022-11-19
> **Newer Backbones**
>
> (Not evaluated over state-of-the-art semantic segmentation architectures)
>
> While we agree with the reviewers that a limitation for broad adoption of our work is its evaluation over older architectures such as FCN-8s and Deeplab-v2, we do believe that this was the right decision for our experimental design as all comparable prior works evaluate over these model architectures. Switching the backbone architecture, would require a re-implementation of all prior work to re-establish baselines, and our concern was that the amount our reimplementation may have hyperparameters which don’t reflect the best possible version of our comparative methods.
>
> (Old Arch vs More Parameters)
> As the siamese architecture in SynthCP shares parameters between both the synthetic and natural image branches, its not straightforward to produce a similar shared siamese architecture where one branch is contains label predictions and the other contains images which should not share that many similarities in early layers.
>
>
> (Dropout and TCP on BDD transfer)
> As the base implementations of these had errors we were unable to generate results in novel settings and thus we include best reported results from prior work.
>
> (Newer Image Synthesis Backbone)
> Again, we agree with reviewers that establishing a new and modern set of baselines for this problem setting is of interest to larger computer vision community, however we believe this outside the scope of our proposed method.
>
> (BDD Evaluation)
> As stated in table 2. we evaluate FCN-8 for the BDD transfer experiment.

---

### Decision · Program_Chairs · 2023-01-20

**Decision:**

Reject

**Justification For Why Not Higher Score:**

- Experiments needs to be consolidated to include more recent backbones and baselines

**Justification For Why Not Lower Score:**

N/A

**Metareview: Summary, Strengths And Weaknesses:**

The paper introduces ErrorAug, a method performing 3 kinds of data augmentations for predicting errors in semantic segmentation. The authors propose to use a two-branch network that takes as input the image and the augmented segmentation mask to perform error prediction. Experiments are conducted on Cityscapes and on BDD for transfer. The paper initially received three borderline reject and one borderline accept recommendation. Although the reviewers fond the approach simple and meaningful, the experimental results had been highly questioned: old segmentation backbone far from state-of-the-art, generation baseline SynthCP based on SPADE (although generation made huge progresses in the last few years), FSNet model from the literature significantly outperforming the proposed approach but not reported, lack of comparison to anomaly detection or uncertainty quantification baselines (e.g. in terms of calibration). The rebuttal did not answer to these concerns, since no new comparison was given and the discussion was avoided. After rebuttal, there was a consensus among active reviewers that the paper should be rejected.

The AC carefully reads the submission. He considers that the approach draws promising research directions, but that the experiments should be consolidated for being published on a top-tier conference as ICLR. Especially, the backbone and baseline comparisons should be updated to show the applicability of the approach with state-of-the-art methods. The big improvement in image generation methods should also be taken into account in the comparison. In addition to the reviewers' recommendations, the AC points out two other baselines to consider, i.e. the extended version of TCP for semantic segmentation using self-training [A], and adversarial attacks as a powerful augmentation technique for error and OOD prediction [B].
Therefore, the AC recommends rejection, but highly encourages the authors to re-submit their work by taking into account reviewer's remarks and advices.

[A] Confidence Estimation via Auxiliary Models. C. Corbière, N. Thome, A. Saporta, T.H. Vu, M. Cord, P. Pérez. T-PAMI'21. \
[B] Triggering Failures: Out-of-Distribution Detection by Learning From Local Adversarial Attacks in Semantic Segmentation. V. Besnier, A. Bursuc, D. Picard, A. Briot. ICCV'21.